# Optimizing User Profiles via Contextual Bandits for Retrieval-Augmented LLM Personalization

## Abstract

Large Language Models (LLMs) excel at general-purpose tasks, but personalizing their responses to individual users remains challenging. Retrieval augmentation offers a lightweight alternative to fine-tuning by conditioning LLMs on user history records, yet existing strategies rely on heuristics (e.g., relevance to the query) that overlook the true contribution of records to personalization. Through a systematic motivation study, we show that (i) relevance does not reliably predict utility, and (ii) utility is non-monotonic across records: the best user profile is not simply the combination of the best individual records, and adding more records can even hurt performance. To address these limitations, we propose PURPLE, a contextual bandit framework that oPtimizes UseR Profiles for Llm pErsonalization. PURPLE operates as a re-ranking layer over candidate records, balancing efficiency with personalization quality. Across nine real-world personalization tasks spanning classification, regression, and short- and long-text generation, PURPLE consistently outperforms strong heuristic and retrieval-augmented baselines, establishing contextual bandit retrieval as a principled and scalable solution for personalized LLMs. Our anonymized code is available at: `https://anonymous.4open.science/r/ICLR-26-PURPLE-A104/`.

## 1 Introduction

Large Language Models (LLMs) have demonstrated remarkable success in various natural language processing tasks, including text generation, question answering, and dialogue systems. As these models are increasingly applied to personalized applications, such as drafting emails on behalf of users, tailoring responses to individuals based on their own preferences has become a crucial challenge. Existing approaches for personalizing LLMs, such as Parameter-Efficient Fine-Tuning (PEFT) (Hu et al., 2022) and Reinforcement Learning from Human Feedback (RLHF) (Ouyang et al., 2022), generally require modifying model parameters. These approaches incur high computational costs, demand frequent updates, and are impractical for real-time personalization at scale, especially when the LLM is not fully open-sourced or the end user cannot afford model fine-tuning. Moreover, continually fine-tuning models for different individuals would complicate safety evaluation and deployment, since each personalized variant would require separate testing.

In this paper, we focus on a lightweight approach for LLM personalization through retrieval augmentation (Wu et al., 2025), where user profiles are constructed by retrieving and injecting a collection of past user records into the prompt to guide personalized responses. Building on this retrieval-augmented view of personalization, prior work has shown that incorporating user profiles can effectively steer LLM outputs toward individual preferences (Salemi et al., 2024; Jiang et al., 2025). Compared to parameter-updating methods, this approach is attractive because it is lightweight, transparent, and readily deployable, since the users can directly inspect and edit the records that guide generation. However, a central challenge remains: **which** user records should be used to form the user profile? Simply appending the entire user history records not only risks introducing redundancy and noise, but can also overflow the model's context window, for example, when histories span years of interactions. Conversely, overly aggressive pruning may discard personalization signals. Existing strategies for building user profiles often rely on heuristics, such as selecting user records with the highest *relevance*, i.e., the similarity to the query (Karpukhin et al., 2020). How-

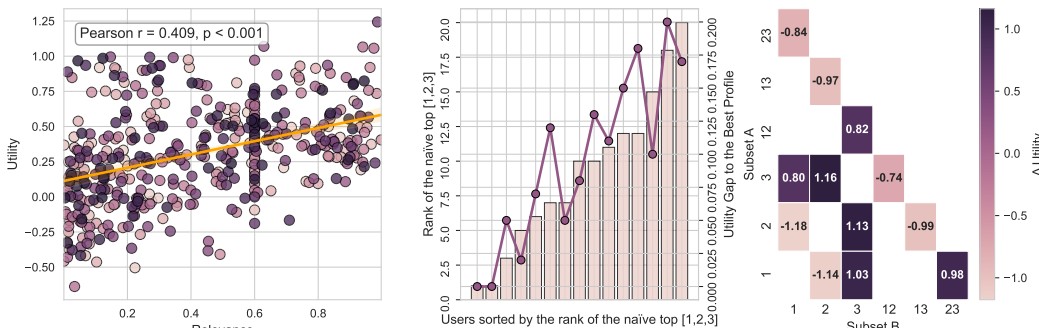

Figure 1: Empirical study of record relevance and personalization utility on *Personalized Product Review Generation* task. (Left) Scatter plot of history records from 15 representative users. Each point is a record, with relevance (semantic similarity to the query) on the $x$-axis and utility (BLEU improvement over no-history baseline) on the $y$-axis. While positively correlated overall ($r = 0.41$, $p < 0.001$), high relevance does not reliably imply high utility. (Middle) For each user, we enumerate all profiles of size $k = 3$ from their top-5 records by individual utility and compare them to the naïve greedy top-3 profile. Bars indicate the rank of the greedy profile (lower is better), and the green line shows its utility gap to the optimal profile. Greedy aggregation often yields suboptimal personalization. (Right) Heatmap of $\Delta$utility for combinations of top-3 records, comparing the joint utility of profile unions with the sum of their parts. Negative values reveal diminishing returns when strong records are combined, while positive values highlight synergies among moderate records. Together, the results show that personalization utility is misaligned with relevance and non-monotonic across records, motivating adaptive selection methods. Full details are provided in Appendix A.

ever, relevance alone does not guarantee personalization gains. What truly matters is the *utility* of the chosen records, i.e., how much they improve downstream task performance when injected into the prompt. To investigate how such a heuristic behaves in real personalization tasks, we conduct experiments on *Personalized Product Review Generation*, drawn from the LongLaMP (Kumar et al., 2024) benchmark. This study (see Figure 1 and Appendix A), reveals two key observations:

- **Utility $\neq$ Relevance**: A record that closely matches the query (high relevance) does not always improve generation quality (utility). Although relevance and utility are often positively correlated, relevance alone is an unreliable predictor of personalization benefit.
- **Utility is Non-monotonic**: Combining the records with the highest individual utility does not necessarily yield the best profile. Greedy aggregation can reduce performance when records overlap or conflict, whereas certain less obvious combinations may provide greater gains.

These two observations highlight what an ideal solution for retrieval-augmented LLM personalization must achieve. At its core, the system needs a re-ranking module that can select a subset of user records whose combined utility is maximized, rather than relying on individual relevance heuristics. To succeed, such a module should satisfy two key requirements. First, its training supervision must come directly from downstream generation quality, not from semantic similarity between the query and records, ensuring alignment with the true personalization objective. Second, it must be list-aware, explicitly modeling dependencies among records so that the selected set captures complementary signals rather than merely aggregating the top individual items. Unfortunately, existing methods fall short: heuristic RAG pipelines satisfy neither requirement, while recent LLM-based list-wise rerankers address dependency modeling but still rely on relevance-oriented supervision.

Motivated by these gaps, we propose **PURPLE**, a framework that models user record selection as a contextual bandit problem (Langford & Zhang, 2007). In its formulation, the context consists of both the current query and the user's past records. The selection policy is guided by a reward function reflecting downstream personalized text generation performance. PURPLE outputs a *propensity score* for each user record, which is passed through a Plackett-Luce ranking model to produce the final selected user records. This formulation enables the model to capture interactions between records and adaptively select those that are most beneficial for personalization. We train PURPLE end-to-end using the policy gradient method (Sutton et al., 1999).

Our main contributions are as follows:

- We demonstrate **using real-world tasks** that relevance and utility are misaligned and that utility is non-monotonic across records, highlighting fundamental limitations of heuristic-based retrieval.

- We introduce **PURPLE**, a framework that casts retrieval-augmented LLM personalization as a contextual bandit problem, adaptively optimizing user profiles beyond static heuristics.

- We show through **extensive experiments** on nine real-world personalization tasks, covering classification, regression, short-text generation, and long-text generation, that PURPLE consistently outperforms strong baselines in both effectiveness and efficiency.

## 2 RELATED WORK

**LLMs for Personalization.** LLMs demonstrate strong performance across domains (OpenAI, 2024), yet their outputs often diverge from user expectations because pre-training captures general rather than individual needs. Reinforcement learning from human feedback (RLHF) (Ouyang et al., 2022) and parameter-efficient finetuning (PEFT) (e.g., LoRA (Hu et al., 2022)) can align models with user preferences, but both require model finetuning and are impractical for end users who lack access or resources. A complementary direction personalizes LLMs through user profiles (Salemi et al., 2024), built from prior user interactions or external signals. Incorporating user profiles into the prompt has shown benefits across multiple tasks requiring personalization, including text summarization (Zhang et al., 2024), question answering (Wu et al., 2024), content generation (Shen et al., 2024), and personalized chatbot interaction (Jiang et al., 2025). Yet it remains unclear which user history records in a profile truly drive performance improvements, particularly in retrieval-augmented generation (RAG), where performance hinges on selecting semantically relevant context. Moreover, little analysis has been conducted on how to best select and compose user records into profiles with high personalization utility. Our work addresses this gap by studying how user profiles shape personalization in retrieval-augmented LLMs, and by proposing strategies for selecting user records to maximize downstream performance.

**Retrieval-Augmented Language Models.** Retrieval-augmented language models (RALMs) enhance parametric LMs with external memory to improve factuality and coverage. Early work such as REALM (Guu et al., 2020) and RAG (Lewis et al., 2020) jointly trained the retriever and LM, while Re2G (Glass et al., 2022) further incorporated a reranking module into this end-to-end pipeline. To reduce training costs, subsequent methods froze the LM and applied retrieval in-context. For example, In-Context RALM (Ram et al., 2023) leveraged LLMs for reranking, while REPLUG (Shi et al., 2024) distilled retrievers from LLMs. More recently, instruction-tuned variants such as Self-RAG (Asai et al., 2024) and RankRAG (Yu et al., 2024) jointly model retrieval and generation, but their reliance on large-scale finetuning renders them impractical for personalization.

The most relevant to our work are In-Context RALM and REPLUG, yet both incorporate only one retrieved record at a time, a limitation our method directly addresses. Specifically, REPLUG combines multiple records by weighting generation outputs with retrieval probabilities, while In-Context RALM periodically triggers retrieval during decoding at fixed steps and replaces previously used records. These designs arise because jointly reasoning over multiple records leads to a combinatorial explosion in the number of possible profiles. In contrast, our approach is explicitly designed to overcome this limitation by modeling cross-record dependencies and directly optimizing over multi-record profiles without resorting to such approximations.

**LLMs for Reranking.** Reranking methods are commonly categorized as pointwise, pairwise, or listwise. Pointwise models such as MonoBERT (Nogueira et al., 2019) and MonoT5 (Nogueira et al., 2020) score each query–document pair independently, while pairwise models such as DuoT5 (Pradeep et al., 2021) compare candidates in pairs. In contrast, listwise approaches jointly model the full candidate set and have recently been advanced by LLMs through prompt-only ranking (RankGPT (Sun et al., 2023)), distillation into smaller models (e.g., RankVicuna, RankZephyr, Lit5Distill, FIRST (Pradeep et al., 2023a;b; Tamber et al., 2023; Gangi Reddy et al., 2024)), and inference-time relevance extraction (ICR (Chen et al., 2025)). However, these methods conflate relevance with utility, which is insufficient for personalization. In this work, we instead train rerankers using downstream generation quality as feedback, prioritizing utility over semantic similarity.

## 3 METHODOLOGY

We formulate retrieval-augmented LLM personalization as a contextual bandit problem (Langford & Zhang, 2007), where the goal is to learn a policy that selects informative user records given the context. Unlike classic multi-armed bandits, the contextual bandit framework incorporates auxiliary information (e.g., the current query and user history) before making a selection. This formulation enables direct optimization of retrieval strategies through policy gradient reinforcement learning, aligning the selection of user records with downstream personalization objectives.

### 3.1 PROBLEM FORMULATION

We consider a dataset $\mathcal{D} = \{(\mathcal{H}^u, x^u, y^u)\}_{u=1}^{|\mathcal{D}|}$, where each example consists of a user's collection of history records $\mathcal{H}^u$, a query $x^u$ to which the system is asked to provide an answer, and a ground-truth personalized response $y^u$. Personalization is achieved by retrieving informative records from $\mathcal{H}^u$ and supplying them as context to a frozen LLM, which then generates the final response. In practice, we apply PURPLE as a re-ranking module on top of a candidate pool selected by lightweight heuristics, ensuring low-latency inference compatible with large-scale systems. In the following development, we focus on a single user and omit the superscript index for brevity.

Let $\mathcal{H} = \{h_1, \ldots, h_N\}$ denote the set of $N$ history records for a user, where each record $h_i = (x_i, y_i)$ is an input–output pair (e.g., a query and its answer from the user). Given a new query $x$, our goal is to construct a user profile from $\mathcal{H}$ to condition the LLM for generating a personalized response. Formally, a profile is an ordered tuple $\mathcal{P} = \langle p_1, \ldots, p_K \rangle \in \mathrm{Perm}_K(\mathcal{H})$, which is a $K$-permutation of $\mathcal{H}$. We stress that the profile is order-sensitive: different permutations of the same $K$ records correspond to distinct profiles and thus provide different inputs to the downstream LLM.

We formulate the selection of $\mathcal{P}$ as a *contextual bandit problem*, where the context is given by the user's history $\mathcal{H}$ and the query $x$, and the action corresponds to selecting $K$ records from $\mathcal{H}$ to construct a profile. Formally, this formulation consists of the following key components:

- **Context:** $\mathcal{C} = (\mathcal{H}, x)$, where $\mathcal{H}$ is the user's collection of history records and $x$ is the query. This representation captures both past user preferences and the immediate intent.
- **Actions:** $\mathcal{P} = \langle p_1, \ldots, p_K \rangle \in \mathrm{Perm}_K(\mathcal{H})$, which corresponds to selecting $K$ distinct records from $\mathcal{H}$ in a particular order. The action thus determines not only which records to use but also how they are arranged. The size of the action space is $N!/(N-K)!$.
- **Reward:** $R(\mathrm{LLM}(\mathcal{P}, x), y)$, a function that measures the quality of the LLM-generated response $\mathrm{LLM}(\mathcal{P}, x)$ relative to the ground-truth personalized response $y$.

We model the policy for selecting user records with a neural distribution $\pi_{\boldsymbol{\theta}}(\cdot \mid \mathcal{C})$, parameterized by $\boldsymbol{\theta}$, which assigns probabilities to candidate user profiles given the context $\mathcal{C}$. The objective is to learn parameters $\boldsymbol{\theta}$ such that the policy assigns higher probabilities to more informative profiles, which ultimately enhance personalized text generation. To this end, we maximize the expected reward over sampled user profiles, optimizing the following objective on a dataset $\mathcal{D}$ spanning multiple users, each associated with a set of history records, a query, and the corresponding ground-truth answer:

$$\mathcal{J}(\boldsymbol{\theta}) = \mathbb{E}_{(\mathcal{H},x,y)\sim\mathcal{D}, \mathcal{P}\sim\pi_{\boldsymbol{\theta}}(\cdot|\mathcal{C})}[R(\mathrm{LLM}(\mathcal{P}, x), y)]. \tag{1}$$

It is challenging to directly optimize Equation 1 since the reward is not differentiable. To address this, we employ the likelihood ratio gradient estimator from reinforcement learning and stochastic optimization (Williams, 1992; Sutton et al., 1999), which allows us to compute the gradient as:

$$\nabla_{\boldsymbol{\theta}}\mathcal{J}(\boldsymbol{\theta}) = \mathbb{E}_{(\mathcal{H},x,y)\sim\mathcal{D}, \mathcal{P}\sim\pi_{\boldsymbol{\theta}}(\cdot|\mathcal{C})}[\nabla_{\boldsymbol{\theta}}\log\pi_{\boldsymbol{\theta}}(\mathcal{P} \mid \mathcal{C})R(\mathrm{LLM}(\mathcal{P}, x), y)]. \tag{2}$$

Since it is intractable to enumerate all profiles $\mathcal{P} \in \mathrm{Perm}_K(\mathcal{H})$ during the optimization process, we estimate Equation 2 by randomly sampling $M = 32$ profiles. To stabilize training and reduce variance in gradient estimation, we apply z-score normalization over the rewards of these 32 profiles sampled for each example. The detailed gradient estimation scheme is provided in Equation 2 in Appendix B.

Figure 2: Workflow of the proposed PURPLE framework. User records encoder takes a user query and a list of user history records as input, outputting the propensity scores of all records. During **training**, a Plackett-Luce model is employed to convert the propensity scores to a probability distribution over all possible profiles, followed by sampling $M$ profiles for gradient estimation. During **inference**, records with top $K$ propensity scores are provided to the LLM along with the user query to generate a personalized response.

## 3.2 MODEL AND FUNCTION DESIGN

**Design of $\pi_{\boldsymbol{\theta}}(\cdot \mid \mathcal{C})$**   Since different permutations of the selected records may lead to different final responses, we adopt the Plackett–Luce (PL) model, which assigns probabilities to profiles based on the scores of individual user records. Therefore, $\pi_{\boldsymbol{\theta}}(\cdot \mid \mathcal{C})$ defines a distribution over all $(N)_K = N!/(N-K)!$ permutations of length $K$ drawn from the $N$ history records. The probability assigned to a specific profile $\mathcal{P}$ is given by:

$$\pi_{\boldsymbol{\theta}}(\mathcal{P} \mid \mathcal{C}) = \prod_{k=1}^{K} \frac{f_{\boldsymbol{\theta}}(p_k; \mathcal{C})}{S - \sum_{j=1}^{k-1} f_{\boldsymbol{\theta}}(p_j; \mathcal{C})}, \tag{3}$$

where $S = \sum_{i=1}^{N} f_{\boldsymbol{\theta}}(h_i; \mathcal{C})$, and $f_{\boldsymbol{\theta}}(\cdot)$ is the user record encoder that outputs a propensity score in $[0, 1]$ for each record, indicating the model's tendency to include that record in the user profile. During training, profiles are generated by sampling $K$ records without replacement based on Equation 3 At inference time, the top-$K$ records ranked by propensity scores are selected to construct the user profile. Because our user record encoder is order-aware and rewards are assigned to ordered sets, the learned propensity score can be interpreted as each record's contribution to the selected set. We further show in Sec. 5.3 that this ordering achieves higher final utility compared with other baselines.

**Design of $f_{\boldsymbol{\theta}}$**   For the record encoder $f_{\boldsymbol{\theta}}$, we aim to capture the interdependencies among user records. A key design consideration is the trade-off between modeling dependencies at the token level versus the sentence level. While the former could, in principle, capture finer-grained interactions, it would quickly exceed the encoder's context length. To address this, we adopt a late interaction strategy (Khattab & Zaharia, 2020), where we first obtain sentence-level embeddings with a pre-trained encoder, and then apply a Transformer encoder to model dependencies across records. Figure 2 illustrates the overall workflow of our method. Within the user record encoder, we utilize a pre-trained Contriever (Izacard et al., 2022) to obtain token embeddings for both the query and the records. Each record first cross-attends to the query at the token level, producing query-fused record embeddings that incorporate query information. A subsequent pooling operation is then applied to the updated record token embeddings to produce fixed-size sentence-level embeddings. embeddings are then processed by a Transformer encoder to model cross-record dependencies. We omit positional encodings to avoid ordering bias among records.

**Design of Reward Function**   In this work, we propose an LLM-driven reward, where the policy is trained to maximize the log-likelihood that the LLM assigns to the target sequence. Formally, given

Table 1: Results of PURPLE and baselines on six datasets from the LaMP benchmark (Salemi et al., 2024). Out-of-memory results are indicated by "–". The best and second-best result in each column is highlighted in **bold** and underlined.

| Task | Citation | Movie | Rating | News | Scholar | Tweet |
|---|---|---|---|---|---|---|
| Metric | Acc. / F1 | Acc. / F1 | MAE / RMSE | RG1 / RGL / MT | RG1 / RGL / MT | RG1 / RGL / MT |
| *With Phi-4-Mini-Instruct (3.84B)* | | | | | | |
| BM25 | 63.3 / 62.9 | 32.7 / 27.8 | 0.444 / 0.860 | 14.2 / 12.6 / 11.8 | 39.8 / 33.2 / 42.3 | 38.3 / 33.5 / 35.2 |
| Contriever | 65.0 / 64.7 | 35.5 / 30.8 | 0.409 / 0.792 | 14.6 / 13.1 / 12.3 | 39.7 / 33.4 / 41.9 | 38.5 / 33.8 / 35.8 |
| IC-RALM-Llama-3-8B-Instruct | 62.2 / 62.1 | 33.5 / 28.8 | 0.460 / 0.836 | 13.4 / 11.8 / 11.0 | 37.5 / 30.8 / 40.6 | 38.3 / 33.5 / 35.4 |
| REPLUG-LSR | 51.8 / 46.3 | 36.8 / 32.6 | 0.498 / 0.913 | 14.1 / 12.7 / 11.5 | 14.1 / 12.7 / 11.5 | **42.3** / **37.3** / **38.9** |
| RankGPT-Llama-3-8B-Instruct | 64.9 / 64.5 | 33.1 / 27.5 | 0.444 / 0.852 | 14.3 / 12.8 / 12.0 | 39.7 / 33.3 / 42.0 | 38.2 / 33.5 / 35.3 |
| RankGPT-GPT5-nano | 65.9 / 65.6 | 35.5 / 31.4 | 0.444 / 0.865 | 14.6 / 13.0 / 12.1 | 39.8 / 33.4 / 42.3 | 38.5 / 33.7 / 35.5 |
| ICR-Llama-3-8B-Instruct | 65.8 / 65.6 | 33.2 / 28.5 | 0.420 / 0.810 | 15.0 / 13.4 / **12.5** | 39.6 / 33.0 / 42.0 | 38.8 / 33.9 / 35.7 |
| PURPLE (Ours) | **66.2** / **65.8** | **38.2** / **33.6** | **0.405** / **0.788** | **15.2** / **13.5** / **12.5** | **40.0** / **33.5** / **42.4** | 39.1 / 34.0 / 35.9 |
| *With Llama-3-8B-Instruct (8.03B)* | | | | | | |
| BM25 | 56.1 / 55.8 | 45.7 / 37.7 | 0.345 / 0.689 | 16.3 / 14.6 / 14.3 | 41.0 / 35.1 / 40.8 | 31.2 / 26.4 / 27.3 |
| Contriever | 58.7 / 58.6 | 46.8 / 38.8 | 0.320 / 0.641 | 17.2 / 15.5 / 15.1 | 41.2 / 35.5 / 40.5 | 31.9 / 26.9 / 28.3 |
| IC-RALM-Llama-3-8B-Instruct | 59.4 / 57.0 | 37.0 / 29.4 | 0.366 / 0.680 | 13.8 / 12.2 / 12.0 | 36.1 / 30.1 / 39.1 | 30.1 / 25.3 / 26.2 |
| REPLUG-LSR | 54.2 / 45.0 | 40.3 / 30.4 | 0.318 / 0.638 | 14.7 / 13.2 / 11.7 | **42.6** / **37.3** / 40.9 | 30.7 / 26.3 / 26.2 |
| RankGPT-Llama-3-8B-Instruct | 56.7 / 56.3 | 46.1 / 37.7 | 0.330 / 0.649 | 16.7 / 15.1 / 14.4 | 41.1 / 35.5 / 40.7 | 31.2 / 26.4 / 27.5 |
| RankGPT-GPT5-nano | 59.5 / 58.0 | 45.1 / 36.2 | 0.321 / 0.638 | 17.1 / 15.4 / 15.0 | 41.0 / 35.3 / 40.5 | 31.5 / 26.5 / 27.8 |
| ICR-Llama-3-8B-Instruct | 58.7 / 57.8 | 47.5 / 38.4 | 0.326 / 0.662 | 17.1 / 15.4 / 14.9 | 41.5 / 35.8 / **41.1** | 31.4 / 26.5 / 27.8 |
| PURPLE (Ours) | **60.2** / **59.8** | **48.8** / **41.0** | **0.316** / **0.637** | **17.7** / **15.9** / **15.4** | 42.0 / 36.7 / 40.8 | **32.6** / **27.5** / **28.8** |
| *With Llama-3-70B-Instruct (70.6B)* | | | | | | |
| BM25 | 70.9 / 70.4 | 54.0 / 46.7 | 0.254 / 0.554 | 17.7 / 16.1 / 14.5 | 43.1 / 37.7 / 39.9 | 36.1 / 30.7 / 32.8 |
| Contriever | 70.2 / 69.9 | 56.4 / 49.1 | 0.240 / 0.530 | 18.5 / 16.9 / 15.5 | 44.2 / 38.8 / 41.1 | 36.5 / 31.4 / 33.3 |
| IC-RALM-Llama-3-8B-Instruct | 66.5 / 66.4 | 49.3 / 41.9 | 0.260 / 0.553 | 14.8 / 13.3 / 12.2 | 39.7 / 34.1 / 38.6 | 32.0 / 27.4 / 28.8 |
| REPLUG-LSR | 66.2 / 65.9 | 51.7 / 43.9 | - / - | 15.2 / 13.8 / 12.1 | 0.0 / 0.0 / 0.0 | 32.2 / 27.8 / 27.8 |
| RankGPT-Llama-3-8B-Instruct | 69.5 / 68.9 | 56.8 / 49.3 | 0.251 / 0.555 | 17.7 / 16.1 / 14.9 | 44.0 / 38.5 / 41.0 | 35.8 / 30.6 / 32.3 |
| RankGPT-GPT5-nano | **73.8** / **73.5** | 55.3 / 48.2 | 0.240 / 0.523 | 18.7 / 17.0 / 15.8 | **44.6** / 38.8 / 41.5 | 36.6 / 31.3 / 33.2 |
| ICR-Llama-3-8B-Instruct | 71.4 / 70.8 | 56.5 / 48.9 | 0.240 / 0.536 | 18.3 / 16.7 / 15.1 | 44.5 / **38.9** / 41.5 | 36.1 / 30.8 / 33.0 |
| PURPLE (Ours) | 72.8 / 72.5 | **57.1** / **50.4** | **0.235** / **0.514** | **18.8** / **17.1** / 15.7 | 44.4 / 38.8 / 41.0 | **37.3** / **32.1** / **34.0** |

a user profile $\mathcal{P}$, a query $x$, and a ground-truth personalized response $y$, we define the reward as:

$$R(\text{LLM}(\mathcal{P}, x), y) = \log p_\phi(y \mid \mathcal{P}, x) = \sum_{j=1}^{|y|} \log p_\phi(y_j \mid \mathcal{P}, x, y_{<j}), \tag{4}$$

where $\phi$ are the parameters of the LLM and $p_\phi(\cdot)$ denotes its next-token distribution. Using the log-likelihood of ground-truth sequences as the reward provides dense feedback signals, in contrast to downstream metrics such as accuracy, mean squared error, or ROUGE-1 (Liu et al., 2025). Moreover, we show in Appendix C that this objective is equivalent to maximizing the evidence lower bound (ELBO) of the marginalization-based RAG approach (Lewis et al., 2020), which, however, becomes intractable in our setting due to the combinatorial explosion. In the next section, we empirically demonstrate that this log-likelihood–based reward is robust across diverse downstream tasks.

## 4 EXPERIMENTS

### 4.1 DATASET AND EVALUATION

We evaluate the performance of PURPLE using `Phi-4-Mini-Instruct` (Microsoft, 2025) and `Llama-3-8B-Instruct` (Team, 2024) as the frozen LLM for response generation, and further scale up to `Llama-3-70B-Instruct` (Team, 2024). Our experiments span a wide range of personalization settings, including personalized classification, regression, and both short- and long-text generation from the LaMP (Salemi et al., 2024) and LongLaMP (Kumar et al., 2024) benchmarks. We follow the prompt templates of Salemi et al. (2024) and Kumar et al. (2024) to incorporate user profiles into the original queries.

Specifically, we evaluate PURPLE on **nine personalization tasks**: two classification tasks — *Personalized Citation Identification* (Citation) and *Personalized Movie Tagging* (Movie), evaluated with Accuracy and F1; one regression task — *Personalized Product Rating* (Rating), evaluated with MAE and RMSE; and six generation tasks, evaluated with ROUGE-1 (RG1), ROUGE-L (RGL) (Lin, 2004), and METEOR (MT) (Banerjee & Lavie, 2005). The generation tasks are further divided into

short-text generation — *Personalized News Headline Generation* (News), *Personalized Scholarly Title Generation* (Scholar), and *Personalized Tweet Paraphrasing* (Tweet) — and long-text generation — *Personalized Abstract Generation* (Abstract), *Personalized Topic Generation* (Topic), and *Personalized Product Review Generation* (Review). In all experiments, we first use Contriever (Izacard et al., 2022) to retrieve the top 20 records as the user history $\mathcal{H}$, and then select 5 of them with different methods to construct the user profile $\mathcal{P}$.

## 4.2 BASELINE METHODS

We focus on the setting where the LLM is kept frozen and no ground-truth profile is available for training the reranker. Therefore, we compare with three categories of prior methods that, likewise, neither fine-tune the LLM nor rely on supervision from ground-truth retrieval results.

The baselines we compare with include **(i) Zero-Shot Rerankers** that apply pre-trained LLMs directly without further fine-tuning. We compare with ICR Chen et al. (2025) and RankGPT Sun et al. (2023). For both methods, we adopt `Llama-3-8B-Instruct` as the reranker LLM. We also report the performance of RankGPT on GPT-5 nano to reflect methods that distill knowledge from the ranking results of state-of-the-art proprietary LLMs (Pradeep et al., 2023a;b; Tamber et al., 2023; Gangi Reddy et al., 2024). **(ii) In-Context Retrieval-Augmented Language Models** that do not fine-tune the LLM. These include REPLUG-LSR Shi et al. (2024) and In-Context RALM Ram et al. (2023). Both methods consider only one record at a time when generating a response. They incorporate multiple records from the user profile either through marginalization (REPLUG-LSR) or through context switching, where reranking is performed multiple times during decoding to swap in new records (In-Context RALM). Additionally, we include **(iii) Efficient Dense and Sparse Retrievers**, applied directly as rerankers. Specifically, we use the dense retriever Contriever Izacard et al. (2022) and the sparse retriever BM25 Robertson & Zaragoza (2009). These methods represent the efficiency-oriented side of the efficiency–performance trade-off. Due to the space limit, we only briefly describe the baseline methods in the main paper. For detailed illustrations of these baselines, please refer to Appendix D.1.

## 5 EXPERIMENT RESULTS

### 5.1 OVERALL PERFORMANCE COMPARISON

Table 1 presents the results of PURPLE and baseline methods on the LaMP benchmark, while Table 2 contains the results on the LongLaMP benchmark. The main findings are as follows:

**PURPLE consistently outperforms strong baselines across LLM scales** Across all tasks and LLMs of varying sizes, PURPLE achieves consistent improvements over existing methods. Compared with Contriever, which is of comparable model size, our learned propensity scores provide more effective ranking signals than raw relevance. Compared with zero-shot rerankers, namely RankGPT and ICR, which use much larger backbone LLMs and incur higher inference cost, PURPLE achieves stronger personalization with a much smaller model, since training with log-probability rewards allows us to better capture the utility of profiles formed by multiple records. Compared with in-context RALMs, namely REPLUG and In-Context RALM, which provide user records one at a time to the LLM and combine multiple records post hoc, our single-stage modeling more effectively captures personalized signals, highlighting the advantage of treating user profiles holistically.

**PURPLE outperforms baselines with high computational throughput.** Figure 3 shows on a representative LaMP dataset that PURPLE outperforms existing methods while be-

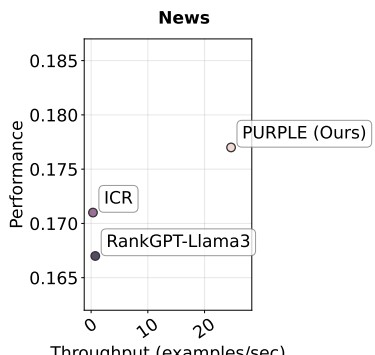

Figure 3: Performance–throughput graph on the News dataset. PURPLE is faster than LLM-based rerankers while achieving better performance.

ing more efficient. We can observe that PURPLE maintains higher performance that ICR and RankGPT-Llama3 while achieving high computational throughput.

**PURPLE is effective across task types, including regression.** Although our reward is based on the log-probability that the LLM assigns to the ground-truth response, it does not directly reflect numerical distances between regression targets, as shown in Table 2. PURPLE still achieves strong gains on regression tasks. This demonstrates that log probability provides a principled and broadly applicable reward signal across diverse task formats.

Table 2: Results of PURPLE and baselines on three datasets from the LongLaMP benchmark (Kumar et al., 2024). The best and second-best result in each column is highlighted in **bold** and underlined.

| Task | Abstract | Topic | Review |
|------|----------|-------|--------|
| Metric | R1 / RL / M | R1 / RL / M | R1 / RL / M |
| *With Phi-4-Mini-Instruct (3.84B)* | | | |
| BM25 | 38.8 / 22.2 / 26.3 | 24.7 / 12.4 / 17.3 | 27.5 / 13.8 / 16.7 |
| Contriever | 38.6 / 21.7 / 26.0 | 23.5 / 12.1 / 16.3 | 27.6 / 13.8 / 16.8 |
| IC-RALM-Llama-3-8B-Instruct | 37.2 / 21.1 / 25.0 | 23.0 / 11.5 / 16.1 | 26.7 / 13.3 / 16.0 |
| REPLUG-LSR | 36.3 / 21.5 / 23.5 | 16.8 / 9.3 / 10.6 | 24.4 / 12.6 / 14.5 |
| RankGPT-Llama-3-8B-Instruct | 38.8 / 22.1 / 26.3 | 24.5 / 12.3 / 17.2 | 27.1 / 13.6 / 16.4 |
| RankGPT-GPT5-nano | **39.1** / **22.4** / **26.9** | **24.9** / **12.5** / **17.5** | 27.1 / 13.7 / 16.6 |
| ICR-Llama-3-8B-Instruct | 38.8 / 22.2 / 26.4 | 23.6 / 12.1 / 16.2 | 27.8 / 13.9 / 17.0 |
| PURPLE (Ours) | 38.9 / 22.3 / 26.5 | 24.8 / 12.4 / 17.3 | **27.9** / **14.0** / **17.1** |
| *With Llama-3-8B-Instruct (8.03B)* | | | |
| BM25 | 42.2 / 24.2 / 31.7 | 28.9 / 14.3 / 20.4 | 33.4 / 16.3 / 21.3 |
| Contriever | 42.0 / 23.9 / 31.4 | 28.9 / **14.6** / 20.0 | 33.1 / 16.2 / 20.8 |
| IC-RALM-Llama-3-8B-Instruct | 39.4 / 21.3 / 29.5 | 26.1 / 12.7 / 17.9 | 31.3 / 14.8 / 19.5 |
| REPLUG-LSR | 38.7 / 21.1 / 28.7 | 21.7 / 11.5 / 13.5 | 18.0 / 9.9 / 10.1 |
| RankGPT-Llama-3-8B-Instruct | 42.3 / 24.3 / 31.8 | **29.1** / 14.5 / **20.6** | 33.5 / 16.4 / **21.4** |
| RankGPT-GPT5-nano | **42.5** / **24.5** / 32.1 | 28.7 / 14.2 / 20.2 | **33.6** / **16.5** / **21.4** |
| ICR-Llama-3-8B-Instruct | 42.2 / 24.1 / 31.7 | 29.0 / 14.4 / 20.5 | 33.1 / 16.2 / 20.8 |
| PURPLE (Ours) | 42.4 / 24.4 / **32.3** | 28.4 / 14.1 / 19.5 | 33.4 / **16.5** / 21.1 |

## 5.2 ABLATION STUDIES

Table 3 presents the ablation studies of PURPLE using `Llama-3-8B-Instruct`. Overall, we examine two key design choices. First, instead of performing token-level cross attention, we test a simplified variant, referring to w/o CA in Table 3, that encodes the entire query into a single embedding and appends it as an extra token to the Transformer encoder. This approach is less effective, indicating that fine-grained token-level interactions between the query and user records are crucial for accurate personalization. Second, we remove the Transformer encoder entirely, referring to w/o RDM in Table 3, resulting in a point-wise scoring model where each record is scored independently. This variant shows the largest performance drop across tasks. While it can still leverage individually informative records, it fails to model dependencies such as redundancy and complementarity among records. In contrast, the full model with the Transformer encoder captures cross-record dependencies, enabling it to identify overlapping information and combine mutually supportive records, thereby achieving better personalization quality.

These results highlight that both token-level cross attention and cross-record dependency modeling are indispensable, validating our design of treating user profiles as structured contexts rather than isolated records.

## 5.3 ANALYSIS: SELECTING TOPK AT INFERENCE

To further examine the quality of the learned propensity scores, we compare our top-5 selection against baselines including ICR, RankGPT, and Contriever. For each example in the test set, we consider the top-5 records proposed by each method and enumerate $5! = 120$ possible orderings. We then randomly sample 5 orderings as controls. As shown in Figure 4, across this expanded

Table 3: Ablation study of PURPLE. We use both `Phi-4-Mini-Instruct` and `Llama-3-8B-Instruct` for the experiment. CA and RDM stand for cross-attention and record dependency modeling, respectively. The former fuses query and record into token-level representations, while the latter explicitly models dependencies among records.

| Task | Citation | Movie | Rating | News | Scholar | Tweet |
|---|---|---|---|---|---|---|
| **Metric** | Acc. / F1 | Acc. / F1 | MAE / RMSE | RG1 / RGL / MT | RG1 / RGL / MT | RG1 / RGL / MT |
| *With Phi-4-Mini-Instruct (3.84B)* | | | | | | |
| PURPLE | **66.2 / 65.8** | **38.2 / 33.6** | **0.405 / 0.788** | **15.2 / 13.5 / 12.5** | **40.0 / 33.5 / 42.4** | **39.1** / 34.0 / 35.9 |
| w/o CA | 64.8 / 64.5 | 35.1 / 29.7 | 0.440 / 0.816 | 14.8 / 13.2 / 12.4 | **40.0 / 33.5** / 42.2 | **39.1 / 34.1** / 36.0 |
| w/o RDM | 61.3 / 60.6 | 35.0 / 31.1 | 0.449 / 0.850 | 14.5 / 12.8 / 11.9 | 39.7 / 33.1 / 41.9 | 39.0 / 34.0 / **36.1** |
| *With Llama-3-8B-Instruct (8.03B)* | | | | | | |
| PURPLE | **60.2 / 59.8** | **48.8 / 41.0** | **0.316 / 0.637** | **17.7 / 15.9 / 15.4** | **42.0 / 36.7 / 40.8** | **32.6** / 27.5 / **28.8** |
| w/o CA | 57.9 / 57.6 | 47.0 / 39.0 | 0.334 / 0.664 | 16.8 / 15.2 / 14.6 | 40.4 / 34.6 / 40.0 | 32.0 / 27.3 / 28.5 |
| w/o RDM | 55.6 / 55.0 | 44.1 / 37.2 | 0.328 / 0.647 | 16.2 / 14.6 / 14.3 | 39.2 / 33.8 / 38.0 | 32.2 / **27.7 / 28.8** |

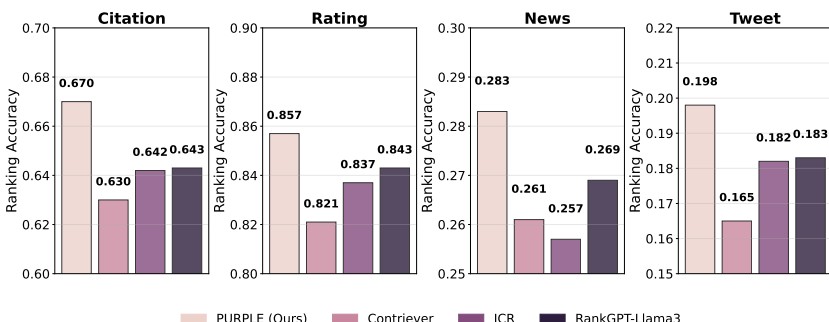

Figure 4: Ranking accuracy comparison across tasks using `LLaMA-3-8B-Instruct`. PURPLE achieves the highest accuracy on all datasets, consistently outperforming heuristic retrievers (Contriever), LLM-based rerankers (RankGPT), and in-context rerankers (ICR).

evaluation, we find that orderings induced by our learned propensity scores are more frequently ranked as the best among the six candidates. This result indicates that our scoring function better captures relative preferences between records, rather than relying on local pairwise relevance alone. These findings highlight that our method not only identifies useful records but also arranges them in an order that maximizes downstream personalization utility.

## 6 CONCLUSION AND DISCUSSION

In this work, we studied the problem of retrieval-augmented personalization for large language models. Through a systematic motivation study, we revealed two fundamental challenges: (i) *record relevance does not reliably predict personalization utility*, and (ii) *utility is non-monotonic* across records, making greedy aggregation suboptimal. To address these limitations, we proposed **PURPLE**, a contextual bandit framework that optimizes user profiles by directly leveraging downstream performance as feedback. PURPLE jointly models query–record interactions and cross-record dependencies, enabling adaptive selection of user profiles beyond static heuristics. Extensive experiments on nine real-world personalization tasks across classification, regression, and text generation showed that PURPLE consistently outperforms heuristic retrievers, LLM-based rerankers, and in-context RALMs, while being significantly more efficient. These results establish contextual bandit retrieval as a principled and scalable paradigm for personalized LLMs. One limitation of our work is that PURPLE requires separate training on each dataset; in future work, we plan to investigate its ability to generalize across tasks and domains. We believe PURPLE opens a promising direction for integrating learning-based profile construction into retrieval-augmented generation, and we hope it inspires future work on reinforcement learning for efficient personalization.

## REPRODUCIBILITY STATEMENT

We place a strong emphasis on reproducibility. In the main text, we provide detailed descriptions of our user record encoder architecture, including the cross-attention mechanism, Transformer encoder design, and the Plackett–Luce formulation for profile selection. In the experiments section, we carefully document the training configurations, such as batch size, learning rate, number of epochs, optimizer settings, and gradient clipping, to facilitate faithful re-implementation. We also release a well-configured codebase that contains scripts for dataset preprocessing, prompt templates, model training, and evaluation. The codebase is designed to be plug-and-play, requiring minimal setup, and ensures that all experiments reported in the paper can be reproduced reliably.

## LLM USAGE STATEMENT

During the preparation of this manuscript, we employed large language models (LLMs) to assist with English writing refinement and style polishing. All technical content, including the design of PURPLE, theoretical formulations, experimental setup, and reported results, was conceived, implemented, and validated by the authors. The LLMs were used solely for linguistic improvement and did not contribute to the research methodology or experimental findings.

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

## A    DETAILS OF EMPIRICAL STUDY

We study the role of user history records in the *Personalized Product Review Generation* task (Kumar et al., 2024). Each data sample corresponds to a unique user with a query $x$, a ground-truth personalized response $y$, and a user history $\mathcal{H}$ consisting of a sequence of (query, response) pairs from the same user. For each record $h_i \in \mathcal{H}$, we measure and define: (i) **Relevance**: semantic similarity between $h_i$ and the query $x$, computed as cosine similarity of their Contriever embeddings: $\mathrm{rel}(h_i) = \cos(\mathrm{Enc}(x), \mathrm{Enc}(h_i))$. (ii) **Utility**: improvement in generation quality when $h_i$ is included in the LLM's prompt. Let $y'$ be the LLM's output without knowing any personalized history records, and $y'_i$ the output with injecting $h_i$ in the prompt. Utility can then be formally defined as the BLEU score improvement by comparing with the ground truth: $\mathrm{util}(h_i) = \mathrm{BLEU}(y'_i, y) - \mathrm{BLEU}(y', y)$. Note that utility can also be extended to a user profile containing multiple records by replacing $h_i$ with a sequence.

**Observ. 1: Utility $\neq$ Relevance.** We first examine whether semantic relevance between a user record and the current query is a reliable proxy for personalization utility. For visualization, we select 15 representative users from the dataset, 5 with relatively few history records, 5 with a medium number, and 5 with a large number, and compute both the relevance score and utility score for each of their records independently. All records are aggregated into a single scatter plot, where different colors denote different users.

Figure 1 (left) plots relevance on the $x$-axis and utility on the $y$-axis. A global regression line (black), along with its 95% confidence interval, is overlaid for easier inspection of the overall trend. The regression indicates a positive correlation (Pearson $r = 0.41$, $p < 0.001$), confirming that relevance and utility are generally related. However, the alignment is far from perfect: many highly relevant records provide little or no utility (points near the bottom-right), while some moderately relevant records deliver large improvements (points near the upper-left). This demonstrates that simply selecting the most relevant records is insufficient for effective personalization, since relevance alone does not reliably indicate utility.

**Observ. 2: Utility is Non-monotonic.** In practice, a user profile generally contains multiple history records. A natural question is whether the best profile (with the highest utility) can be constructed by simply concatenating the individually strongest records, or whether record interactions play a significant role. To test this, we fix the profile size to $k = 3$ and enumerate all $A_5^3 = 60$ ordered profiles formed from the top-5 records of each user, based on their individual utility. We conduct this study on the same 15 users as in Figure 1 (left).

As shown in Figure 1 (middle), the $x$-axis indexes users, sorted by the rank of their naïve top-3 profile (formed by greedily selecting the three records with the highest individual utility). The plot contains two $y$-axes: the left axis (bars) shows the rank position of the naïve profile among all 60 possible profiles, while the right axis (green line) shows the utility gap between the naïve profile and the optimal one. A higher bar means the naïve profile is far from the top-ranked profile, and a larger utility gap indicates the greedy strategy performs substantially worse than optimal. Across users, the naïve top-3 profile rarely achieves the best rank and often incurs a non-trivial utility gap, indicating that greedy aggregation even by utility is frequently suboptimal.

To further illustrate the interaction of user records, Figure 1 (right) examines user profiles of up to three records, drawn from the same top-3 records (again ranked by individual utility). For simplicity, we ignore the order in each profile and analyze interactions between disjoint profiles. Specifically, for two profiles $\mathcal{P}_A$ and $\mathcal{P}_B$, we compute $\Delta\mathrm{util}(\mathcal{P}_A, \mathcal{P}_B) = \mathrm{util}(\mathcal{P}_A \cup \mathcal{P}_B) - \big(\mathrm{util}(\mathcal{P}_A) + \mathrm{util}(\mathcal{P}_B)\big)$, which measures whether the joint utility of the union exceeds (positive) or falls short of (negative) the sum of its components. The heatmap indexes subsets on both axes and each cell reports the average $\Delta\mathrm{util}$ across the 15 users. Cells with overlapping subsets are omitted, since their union would not meaningfully isolate interaction effects. This visualization shows that even when individual or pairwise profiles appear useful, adding them together can reduce overall utility, while certain combinations of moderate records can yield positive gains. Such non-monotonicity underscores that effective user profiles cannot be built by greedily aggregating individually strong records, but must explicitly account for cross-record interactions.

## B   DETAILS OF GRADIENT ESTIMATION

To estimate the gradient in Equation 2, we first draw a batch of examples $\{(\mathcal{H}_b, x_b, y_b)\}_{b=1}^B$. For each example, we sample $M$ profiles $\mathcal{P}_b^1, \ldots, \mathcal{P}_b^M$ from $\pi_\theta(\cdot \mid \mathcal{C}_b)$, and finally compute the empirical mean. This learning procedure corresponds to the REINFORCE algorithm (Sutton et al., 1999), with gradient estimate:

$$\nabla_\theta \mathcal{J}(\theta) \approx \frac{1}{B} \sum_{b=1}^B \frac{1}{M} \sum_{m=1}^M \nabla_\theta \log \pi_\theta(\mathcal{P}_b^m \mid \mathcal{C}_b) \tilde{r}_b^m. \tag{5}$$

To reduce variance in gradient estimation, we apply reward normalization over the $M$ profiles sampled for each example. Concretely, for each example with rewards $\boldsymbol{r}_b = [r_b^1, \ldots, r_b^M]^\top$, where $r_b^m = R(\Phi(\mathcal{P}_b^m, x_b), y_b)$, the normalized reward is computed as $\tilde{r}_b^m = \frac{r_b^m - \text{mean}(\boldsymbol{r}_b)}{\text{std}(\boldsymbol{r}_b)}$.

## C   MOTIVATING OUR REWARD

The specific choice of using the log probability of ground truth personalized response is grounded in the generative modeling perspective of retrieval-augmented generation (RAG) (Lewis et al., 2020), where the user profile is treated as a latent variable and the response likelihood is obtained by marginalizing over all possible profile selections. Applying Jensen's inequality to the training objective in this setting gives:

$$\mathbb{E}_{(\mathcal{H}, x, y) \sim \mathcal{D}} \left[ \log \left( \sum_{\mathcal{P} \in \text{Perm}_K(\mathcal{H})} \pi_\theta(\mathcal{P} \mid \mathcal{C}) p_\Phi(y \mid \mathcal{P}, x) \right) \right] \\ \geq \mathbb{E}_{(\mathcal{H}, x, y) \sim \mathcal{D}, \mathcal{P} \sim \pi_\theta(\cdot | \mathcal{C})} [\log p_\Phi(y \mid \mathcal{P}, x)]. \tag{6}$$

Therefore, maximizing the expected reward under our reinforcement learning objective is equivalent to maximizing the evidence lower bound (ELBO), with $p_\Phi$ modeled by a frozen LLM.

## D   EXPERIMENTAL SETUP

### D.1   DETAILED BASELINE METHODS

We focus on the setting where the LLM is kept frozen and no ground-truth profile is available for training the reranker. This setting is reasonable for personalization as it represents cases where the retrieval corpus consists of past user records and no additional labeling on golden retrieval is required. Therefore, we compare with three categories of prior methods that, likewise, neither fine-tune the LLM nor rely on supervision from ground-truth retrieval results.

The baselines we compare with include **(i) Zero-Shot Rerankers** that apply pre-trained LLMs directly without further fine-tuning. We compare with ICR Chen et al. (2025), which leverages the LLM's attention scores to rank user records, as well as RankGPT Sun et al. (2023), which prompts the LLM to directly output a ranking order. For both methods, we adopt `Llama-3-8B-Instruct` as the reranker LLM, as larger models would incur prohibitive costs in the retrieval pipeline. In addition, there exists a line of rerankers that do not rely on ground-truth supervision but instead distill knowledge from the ranking results of state-of-the-art proprietary LLMs (Pradeep et al., 2023a;b; Tamber et al., 2023; Gangi Reddy et al., 2024). We therefore report the performance of RankGPT with GPT-5 nano to reflect an upper bound of such methods. **(ii) In-Context Retrieval-Augmented Language Models** that do not fine-tune the LLM. These include REPLUG-LSR Shi et al. (2024), which trains the reranker to match the LM likelihood of each user record, as well as In-Context RALM Ram et al. (2023), which leverages the likelihood of recently generated tokens to rerank user records. Both methods consider only one record at a time when generating a response. They incorporate multiple records from the user profile either through marginalization (REPLUG-LSR) or through context switching, where reranking is performed multiple times during decoding to swap in new records (In-Context RALM). These design choices arise because directly evaluating all combinations of records would be computationally intractable under their frameworks, which is a limitation our method aims to overcome. Additionally, we include **(iii) Efficient Dense and Sparse Retrievers**, applied directly as rerankers. Specifically, we use the dense retriever Contriever Izacard

et al. (2022) and the sparse retriever BM25 Robertson & Zaragoza (2009). These methods represent the efficiency-oriented side of the efficiency–performance trade-off.

## D.2 IMPLEMENTATION DETAILS

We employ a frozen pre-trained Contriever to first encode both queries and user history records into token embeddings. The only trainable components are the remaining modules of the user record encoder. These include a cross-attention layer that integrates query information into record embeddings, a Transformer encoder that captures inter-record dependencies, and an MLP decoder that maps the updated record encodings into scalar propensity scores. We set the number of Transformer encoder layers to $l = 12$, resulting in a parameter size roughly twice that of Contriever, while still being substantially faster than the baseline ZSRs and In-Context RALMs. For gradient estimation, we use a batch size of $B = 16$ and sample $M = 32$ user profiles for each example. We train the model for 10 epochs using the Adam optimizer (Kingma & Ba, 2017) with $\beta_1 = 0.9$, $\beta_2 = 0.999$, and a learning rate of $1 \times 10^{-4}$. During training, we apply a gradient clipping norm of 1.0. The checkpoint achieving the best validation performance is selected for testing.

In all experiments, we use frozen LLMs both to generate personalized responses and to evaluate the log probability of ground-truth responses conditioned on the query and user profiles (i.e., our reward). For generations, we set the temperature to $T = 0.7$ and employ nucleus sampling (Holtzman et al., 2020) with $top\_p = 0.8$. For `Phi-4-Mini-Instruct` and `Llama-3-8B-Instruct`, we deploy on a single NVIDIA H100 GPU. For `Llama-3-70B-Instruct`, we deploy the model across four NVIDIA H100 GPUs using vLLM (Kwon et al., 2023). All LLMs are deployed in BF16 precision. Training of PURPLE is conducted on the same GPUs used for LLM deployment.

