# OpenReview forum: "Optimizing User Profiles via Contextual Bandits for Retrieval-Augmented LLM Personalization"
_ICLR.cc/2026/Conference — ICLR 2026 Conference Withdrawn Submission_

### Official Review · Reviewer_pkZa · 2025-10-31

**Soundness:** 3
**Presentation:** 2
**Contribution:** 2
**Rating:** 4
**Confidence:** 3

**Summary:**

The paper studies personalization for LLMs when conditioning on user history. It argues that common heuristic retrieval (e.g., relevance to the query) does not faithfully capture the true utility of records for personalization, and that utility across records is non-monotonic—adding more “good” records can hurt. The authors propose PURPLE, a contextual bandit–based re-ranking layer over candidate user records that aims to optimize user profiles for downstream tasks. They report improvements over heuristic and retrieval-augmented baselines across classification, regression, and short/long text generation tasks, and position contextual bandit retrieval as a scalable path to personalized LLMs.

**Strengths:**

- Framing RAG-based personalization as a contextual bandit problem is an interesting and potentially impactful idea.
- The paper includes experiments on multiple datasets, aiming to demonstrate applicability across task types.

**Weaknesses:**

1. The central empirical claims lack key analyses and breadth: despite the broader narrative, the core observations are effectively demonstrated on **a single model and a single dataset**, leaving generality unsubstantiated and the evidence shallow
2. The claim that relevance ≠ utility lacks deeper analysis: the paper does not explain when/why alignment breaks, nor provide theoretical grounding for the observed non-monotonicity (e.g., conditions under which relevant records become non-useful or harmful).
3. Baseline coverage is incomplete. Important personalized RAG methods are missing, such as [1,2]. In addition, comparisons against commonly used PAG methods are absent.

[1] Retrieval Augmented Generation with Collaborative Filtering for Personalized Text Generation (SIGIR 2025)

[2] Measuring What Makes You Unique: Difference-Aware User Modeling for Enhancing LLM Personalization (ACL Findings 2025)

**Questions:**

- The statement that “combining the records with the highest individual utility does not necessarily yield the best profile.” Why does PURPLE resolve this? Please clarify the mechanism or assumptions under which the proposed method overcomes this combinatorial non-monotonicity.

- Whether there are concrete failure cases where “relevance alone” is a poor predictor of personalization benefit? What characteristics of the query/user/history drive these failures?

- Is there any theoretical support for the key claims (e.g., relevance–utility misalignment, non-monotonicity), even under simplifying assumptions?

---

> ### Author Response · Authors · 2025-12-03
>
> Thank you for your thorough review and your appreciation of the merits of our method. We address your main questions below.
>
> **W1: Breadth of Empirical Study**
>
> As an empirical/motivation study, our visualization focuses on one representative dataset to isolate the two phenomena (relevance–utility misalignment and non-monotonicity). However, their generality is supported by our results across nine tasks, multiple LLM scales, and two benchmarks (Tables 1–2).The consistent improvements of PURPLE over relevance-based and LLM-based rerankers across classification, regression, and short/long text generation indicate that these effects are not dataset/model-specific.
>
>
> **Q2: Concrete Example(s) of Relevance–alone Failures**
> A core limitation of relevance-only retrieval is that semantic similarity emphasizes surface context rather than what actually benefits personalization. Because each past user record mixes contextual phrasing (e.g., “Friday night,” “recommend a movie”) with deeper preference content (e.g., liking thrillers or comedies), a relevance scorer often ranks records highly simply because their wording resembles the new query. However, personalization utility depends on whether the underlying preference signal meaningfully supports the user’s current intent, not mrely contextual overlap. As a result, relevance-alone can prioritize misleading records that harm personalization utility. We provide a concrete example below.
>
>
> Consider a user launches a new query **"Recommend something relaxing for Friday night."** and three past user records (each with a query and a response lets say)
> >
> > **h₁ (contextually similar but preference-misaligned)**
> >
> > query: “Recommend a movie for a Friday night when I want something intense.”
> >
> > response: “I enjoy dark psychological thrillers with twists.”
> >
>
> > **h₂ (moderate contextual similarity, preference-aligned)**
> >
> > query: “Give me ideas for movies when I want to unwind.”
> >
> > response: “I enjoy lighthearted romantic comedies that help me relax.”
>
> > **h₃ (moderate contextual similarity, preference-aligned)**
> >
> > query: “Suggest a film that will cheer me up after a long day.”
> >
> > response: “I like uplifting musicals that make me feel good.”
>
>
> As we can see, record h1 is highly relevant to the current new query due to strong contextual overlap (“Friday night,” “recommend a movie”), yet its preference content contradicts the user’s current intent and thus yields low or even negative utility. In contrast, records h2 and h3 (i.e., past requests about movies “to unwind” or films “to cheer me up,” with responses favoring lighthearted comedies or uplifting musicals) have only moderate contextual similarity but encode preference signals that align directly with the user’s intent, resulting in substantially higher personalization utility. This illustrates that high relevance alone between the query-record can mislead, while lower-relevance records may provide stronger utility, matching the patterns observed in Fig. 1.

---

> ### Author Response · Authors · 2025-12-03
>
> **W2/Q3: More Support and Theoretical Grounding for Core Claims**
>
> We appreciate the reviewer’s request for more support and theoretical grounding. Our current version emphasizes empirical characterization (Fig. 1), but the phenomena we observe can be understood more structurally. At a high level, each record h_i can be seen as combining contextual phrasing (how similar it sounds to the current query) and preference content (what it reveals about the user). Relevance scores
> rel(h_i, x) are computed as semantic similarity between the new query x and the entire record text, which tends to emphasize context overlap, while personalization utility util(h_i, x) measures the improvement in downstream performance when the LLM conditions on that record.
>
>
> **[When relevance-utility alignment breaks?]**
> Alignment breaks precisely when these two components pull in different directions: for example, in the movie scenario above, a past record about “Friday night” and “movie recommendations” is highly relevant at the text level but encodes a preference for intense thrillers, which is harmful for the new intent “something relaxing.” More generally, misalignment is most pronounced for (i) multi-interest users, where query-specific intent only matches a subset of the user’s long-term tastes; (ii) contextual mismatch, where the surface form of the query (e.g., “Friday night,” “after a long day”) resembles older queries whose underlying preferences are different; and (iii) redundant or conflicting records, where repeated or contradictory signals either saturate or confuse the LLM. These patterns are exactly what we see in Fig. 1 (left): many high-relevance records live near zero or negative utility, while moderately relevant records that carry the “right” preference cues provide the largest gains. In the revision, we will explicitly add this breakdown of when/why alignment breaks, together with the concrete case study example.
>
> **[Support for non-monotonicity of utility]**
> For non-monotonicity and theoretical support, our empirical analysis in Fig.~1 (middle/right)
> suggests that \emph{utility over sets of records is not additive}: the joint utility of a profile
> $P$ is not simply the sum of its parts. A natural way to express this is:
>
> $$\text{util}(P)=\sum_{i \in P} u_i\;+\;\sum_{\substack{i < j \\ i,j \in P}} \gamma_{ij}$$
>
> where $u_i$ denotes the individual contribution of record $h_i$ and $\gamma_{ij}$ captures
> *interaction effects* between records. Negative interactions ($\gamma_{ij} < 0$) correspond
> to redundancy or interference (diminishing returns), while positive interactions ($\gamma_{ij} > 0$)
> represent complementary preference signals (synergies).
>
> Under this interaction model, it is immediate that:
> (i) selecting the top-$k$ records by individual utility $u_i$ need not maximize $\text{util}(P)$, and
> (ii) utility is in general \textbf{non-monotone}: adding an individually strong record can \emph{decrease}
> overall utility whenever $\gamma_{ij} < 0$ for some existing records.
>
> This aligns with the heatmap in Fig.~1 (right), where combining individually strong records yields
> negative $\Delta$utility, whereas mixtures of moderate records produce positive synergies.
>
>
> Our reward definition also has a formal grounding. As shown in Appendix C, the record-level reward—log-likelihood of the ground-truth personalized answer—is equivalent to maximizing the ELBO of a marginalization-based retrieval objective. This probabilistic interpretation further supports our treatment of utility and explains why relevance (a similarity measure) is not aligned with optimization of downstream task likelihood. In the revision, we will incorporate this interaction-based explanation, clarify its implications for non-monotonicity, and move the ELBO connection more prominently into the main analysis.

---

### Official Review · Reviewer_z1Zm · 2025-11-01

**Soundness:** 2
**Presentation:** 3
**Contribution:** 2
**Rating:** 4
**Confidence:** 4

**Summary:**

This paper primarily explores how to construct user profiles more efficiently in large language model (LLM) personalization. The authors point out that traditional retrieval augmentation (RAG)-based personalization methods typically rely on "relevance to the query" to select user history records. However, this heuristic approach has two fundamental problems: ① relevance does not equal true personalization utility; ② the utility between different records is non-monotonic, and simply stacking the most relevant records may actually reduce performance. To address these issues, the paper proposes the PURPLE framework ("oPtimizing UseR ProfiLes for llm pErsonalization"), which models user record selection as a contextual bandit problem. Through policy gradient reinforcement learning optimization, it directly uses the performance of downstream generation tasks as reward signals, dynamically learning which record combinations best improve personalization results. Experiments on nine real-world tasks (classification, regression, short text generation, and long text generation) demonstrate that PURPLE significantly outperforms traditional heuristic retrieval, zero-shot re-rankers (such as RankGPT and ICR), and RAG variants (such as REPLUG and In-Context RALM) in both accuracy and efficiency. Research shows that context-based bandit-based retrieval optimization is a more scalable, interpretable, and efficient personalized LLM solution.

**Strengths:**

1. This paper is well-written, and the figures are well-drawn.
2. The model itself is lightweight, which is significant for personalized LLM research.

**Weaknesses:**

1. The lack of comparison and discussion with personalized LLM methods, such as in brackets [], makes the experimental results less convincing.
2. There is no further discussion on the method's design space, such as why this problem cannot be solved based on RL like [2].
3. The model's performance and application are not discussed for new users or changes in user interests.
4. The method has limited innovation. There are many previous methods for personalized recommendations using context bandit, and further discussion is needed on the differences and connections between these methods.

[1] LaMP: When Large Language Models Meet Personalization

[2] PREMIUM: LLM Personalization with Individual-level Preference Feedback

[3] Personalized Pieces: Efficient Personalized Large Language Models through Collaborative Efforts

[4] Personalized Language Modeling from Personalized Human Feedback

**Questions:**

Please see the weaknesses.

---

> ### Author Response · Authors · 2025-12-03
>
> Thank you for your thorough review and your appreciation of the merits of our method. We address your main questions below.
>
> **W1&W2: Comparison and discussion with other personalized LLM methods**
>
> [1] proposes two LLM personalization strategies: In-Prompt Augmentation (IPA) and Fusion-in-Decoder (FiD). IPA represents the standard processing we apply to our method as well as all compared baselines: integrate all retrieved records into a single prompt and feed it to a frozen LLM. FiD relies on encoder-decoder models as it encodes multiple inputs separately within the encoder; therefore, it also requires further fine-tuning of the LLM, which our method aims to avoid. For a fair comparison, we do not compare with methods that require further fine-tuning of LLMs and only compare different retrieval/reranking methods.
>
> [2] relies on a pre-defined Tag Library to build candidate tag sets based on user query using a neural network. The network is trained by leveraging users' preference feedback on LLM responses generated from different candidate tag sets. While a tag set can be easily identified for certain personalization tasks like personalized movie tagging (LaMP-2), this is not the case for general personalized generation tasks. Moreover, the tag sets considered in this work are static topical features (e.g., Investment, Bakery, Comedy), whereas personalization may involve more intricate attributes such as writing style, tone, and language use. These features are difficult to define using a fixed set of tags but are more explicitly reflected in user history records. We consider a more general setting where only a set of history records (and no online user feedback) is available, and we aim to select a useful subset for retrieval-augmented personalized generation. We do not include PREMIUM in our experiment because we do not rely on pre-defined tag sets or online user feedback on the generated responses.
>
> [3] trains LoRA adaptors for a set of representative users and combine them for new users based on the similarity of the user's profile to existing users, while [4] adapts DPO to incorporate user information and fine-tunes the LLM with the modified DPO. Both of the methods require fine-tuning of the LM, and thus does not belong to the RAG-based methods we compare with.
>
> [1] Salemi et al. "LaMP: When Large Language Models Meet Personalization." ACL, 2024.
> [2] Sun et al. "PREMIUM: LLM Personalization with Individual-level Preference Feedback." OpenReview, 2025.
> [3] Tan et al. "Personalized Pieces: Efficient Personalized Large Language Models through Collaborative Efforts." EMNLP, 2024.
> [4] Li et al. "Personalized Language Modeling from Personalized Human Feedback." Arxiv, 2024.
>
> **W3: New users and changes in user interests**
>
> In our experimental setting, the train/validation/test splits represent different users since we use the user-based split of LaMP and LongLaMP. Therefore, our results represent generalization performance on new users.

---

### Official Review · Reviewer_gsBw · 2025-11-01

**Soundness:** 2
**Presentation:** 3
**Contribution:** 2
**Rating:** 2
**Confidence:** 4

**Summary:**

In this paper, the author proposes Purple, a light-weight retrieval ranking algorithm for personalization. It pivots on the intuition that the retrieved user profiles will not be entirely useful for the language models, and with an effective ranking algorithm, the LLM can get better context when facing the generation task. Purple utilizes a contextual bandit framework that learns to select and rank user history records by optimizing directly for downstream task performance. The empirical results show moderate improvement over classic personalization datasets such as LaMP and LongLAMP.

**Strengths:**

S1. Overall, the paper is clearly written, and the notation is clear. The idea of effectively leveraging the retrieved content is a crucial yet under-explored topic. Current frameworks focus on retrieving context and ignores the idiosyncrasies of the retrieved context.

S2. The paper also present interesting findings in Figure 1 where the empirical results show mixed results on the inclusion of the retrieved content, which gives crucial insights to future researchers.

S3. The problem formulation and method is technically sound and the idea of modeling the task of personalization as a contextual bandit optimization problem is novel.

**Weaknesses:**

W1. Although the retrieved results might have different utility, the motivation is not well founded. Current language models are evolving towards increasingly better utilization of longer context [1]. Often the user profile does not exceed the context window limit, even for datasets such as LongLamp. The author only selects top 3 or top 5 user history, which can be arbitrary and not realistic.

W2. The result improvement is marginal. In table 1, multiple results only have around 0.02 performance comparing to basic baselines. For Ratings improvement, the improvement is smaller as Phi-4-Mini demonstrates 0.004, Llama3-8b demonstrates 0.002, and Llama3-70-B demonstrates 0.005 improvement over baseline. Additionally, the performance improvement is even less consistent in the text generation task, as shown in the LaMP dataset (News, Scholar, Tweet) and LongLaMP dataset (Abstract, Topic, Review), with multiple baselines outperforming the proposed PURPLE framework. With the marginal improvement, the author should at least include standard deviation in the experiment reports.

W3. More ablation study is needed. Although the framework is well justified from the design perspective, the specific component needs to be further studies. For example, the user can further explore the different effects of various encoders in the study.

W4. Prior study has shown that semantic metrics alone cannot fully capture personalization [4]. We should include more advanced LLM-as-a-Judge evaluation [4] in the evaluation, as well as case studies on the generated text.

**Questions:**

1.	To address weakness #1, the author could show results from longer context window. For example, investigating if setting k=15 can still maintain the performance improvement from PURPLE.

2.	The author needs to further refine the framework and improve the marginal performance gain, as well as provide additional standard deviation measurement in the experiment section.

3.	Missing citations [2, 3, 4].

4.	Please include LLM-as-a-Judge metrics as it has shown that semantic metrics alone cannot fully capture personalization.

5.	Personalization is highly subjective, it would be helpful if the author can provide extensive case studies in the experiment section.

[1] A Comprehensive Survey on Long Context Language Modeling. Liu, et al.
[2] A Personalized Conversational Benchmark: Towards Simulating Personalized Conversations. Li, et al.
[3] ExPerT: Effective and Explainable Evaluation of Personalized Long-Form Text Generation. Salemi, et al.
[4] Personalized Language Modeling from Personalized Human Feedback. Li, et al.

---

> ### Author Response · Authors · 2025-12-03
>
> Thank you for your thorough review and for appreciating the novelty of our work. We address your main questions below.
>
> **W1: Motivation for selecting the top-5 records**
>
> While current LLMs have greatly expanded their context length, it is well-studied in the Retrieval-Augmented Generation (RAG) literature that including more than five retrieved passages often yields only marginal gains or even harms performance due to the additional noise introduced (see Figure 6 of [1], Figure 8 of [2], and Figure 3 of [3]). In the context of long context language models (LCLMs), retrieval is often applied as a filtering step to avoid the heavy computational cost of processing long contexts [5]. Therefore, selecting a subset of approximately five passages from a larger candidate set is widely adopted in recent work [1–4]. We believe these observations from general RAG settings also hold for personalization, as the benchmark paper [6] advocates for retrieval augmentation with a similarly small scale of 1–4 records "given the inherent context length constraints of many LLMs and the cost of processing long sequences," and notes that "not all entries within a user profile are necessarily relevant to the specific input at hand."
>
> Additionally, we purposefully limit the number of retrieved records so the system can provide better attribution to the user regarding which records are used.
>
> [1] Yu et al. "RankRAG: Unifying Context Ranking with Retrieval-Augmented Generation in LLMs." NeurIPS, 2024.
> [2] Ram et al. "In-Context Retrieval-Augmented Language Models." TACL, 2023.
> [3] Lewis et al. "Retrieval-Augmented Generation for Knowledge-Intensive NLP Tasks." NeurIPS, 2020.
> [4] Glass et al. "Re2G: Retrieve, Rerank, Generate." NAACL, 2022.
> [5] Jiang et al. "LongRAG: Enhancing Retrieval-Augmented Generation with Long-context LLMs." arXiv, 2024.
> [6] Salemi et al. "LaMP: When Large Language Models Meet Personalization." ACL, 2024.
>
> **W2: Regarding the magnitude of improvements**
>
> While we acknowledge that the improvements are modest, we would like to draw attention to the results presented in the benchmark papers [1,2] and related work [3], all of which evaluate retrieval strategies without fine-tuning the LLM. These studies consistently report similarly small performance gaps across different retrieval methods. This pattern suggests that the limited performance spread is not unique to our method but rather reflects a characteristic of the datasets.
>
> [1] Salemi et al. "LaMP: When Large Language Models Meet Personalization." ACL, 2024.
> [2] Kumar et al. "LongLaMP: A Benchmark for Personalized Long-form Text Generation." arXiv, 2024.
> [3] Shi et al. "Retrieval Augmented Generation with Collaborative Filtering for Personalized Text Generation." SIGIR, 2025.

---

### Note · Authors · 2026-01-06

I have read and agree with the venue's withdrawal policy on behalf of myself and my co-authors.